# Conversion of Charge Carrier Polarity in MoTe_2_ Field Effect Transistor via Laser Doping

**DOI:** 10.3390/nano13101700

**Published:** 2023-05-22

**Authors:** Hanul Kim, Inayat Uddin, Kenji Watanabe, Takashi Taniguchi, Dongmok Whang, Gil-Ho Kim

**Affiliations:** 1Sungkyunkwan Advanced Institute of Nanotechnology (SAINT), Sungkyunkwan University (SKKU), Suwon 16419, Republic of Korea; hanulk@skku.edu; 2Department of Electrical and Computer Engineering, Sungkyunkwan University (SKKU), Suwon 16419, Republic of Korea; inayatuddin@skku.edu; 3Research Center for Functional Materials, National Institute for Materials Science, 1-1 Namiki, Tsukuba 305-0044, Japan; watanabe.kenji.aml@nims.go.jp; 4International Center for Material Nanoarchitectonics, National Institute for Materials Science, 1-1 Namiki, Tsukuba 305-0044, Japan; taniguchi.takashi@nims.go.jp; 5Department of Advanced Materials Science and Engineering, Sungkyunkwan University (SKKU), Suwon 16419, Republic of Korea

**Keywords:** 2D material, laser doping, CMOS inverter, FET, MoTe_2_

## Abstract

A two-dimensional (2D) atomic crystalline transition metal dichalcogenides has shown immense features, aiming for future nanoelectronic devices comparable to conventional silicon (Si). 2D molybdenum ditelluride (MoTe_2_) has a small bandgap, appears close to that of Si, and is more favorable than other typical 2D semiconductors. In this study, we demonstrate laser-induced p-type doping in a selective region of n-type semiconducting MoTe_2_ field effect transistors (FET) with an advance in using the hexagonal boron nitride as passivation layer from protecting the structure phase change from laser doping. A single nanoflake MoTe_2_-based FET, exhibiting initial n-type and converting to p-type in clear four-step doping, changing charge transport behavior in a selective surface region by laser doping. The device shows high electron mobility of about 23.4 cm^2^V^−1^s^−1^ in an intrinsic n-type channel and hole mobility of about 0.61 cm^2^V^−1^s^−1^ with a high on/off ratio. The device was measured in the range of temperature 77–300 K to observe the consistency of the MoTe_2_-based FET in intrinsic and laser-dopped region. In addition, we measured the device as a complementary metal–oxide–semiconductor (CMOS) inverter by switching the charge-carrier polarity of the MoTe_2_ FET. This fabrication process of selective laser doping can potentially be used for larger-scale MoTe_2_ CMOS circuit applications.

## 1. Introduction

A material class called transition metal dichalcogenides (TMD) is attracting intense research attention because of their exceptional physical and chemical properties, including thickness-dependent channel electronic properties, free dangling bond surfaces, enormous magnetoresistance, and excellent intrinsic carrier mobility [1,2,3,4]. The flexibility of TMD allows them to be stacked in different configurations to fabricate various devices, allowing future applications to modify their physical properties as needed [5,6]. Thus, their use has been growing in multiple applications in electronic and optoelectronic applications, such as field-effect transistors (FET), inverters, solar cells, and photodiodes [7,8,9,10]. A two-dimensional (2D) bulk material has weak van der Waals bonding between adjacent layers, so it can easily be confined to a monolayer, bi-layer, tri-layer, or multilayer structure using adhesive tape [1,8,11].

There are several TMD materials with small bandgaps. Among them, molybdenum ditelluride (MoTe_2_) has a 1.1 eV monolayer bandgap and an indirect 0.85 eV multilayer bandgap, which is more comparable to the Si bandgap than other TMD materials [12,13,14]. In addition, MoTe_2_ shows intriguing properties of superconductivity, metallicity, and semiconducting [15,16,17]. The polarity of charge carriers can be controlled by using the appropriate metal contact work function in a few MoTe_2_ layers because of its weak Fermi level pinning and small bandgap [18]. It is easier to fabricate multilayer MoTe_2_ at a large scale than few-layer MoTe_2_ exfoliated at a smaller scale. As a result, multilayer MoTe_2_ is a more appropriate material for future electronic circuits. A metal-oxide-semiconductor (CMOS) inverter is a well-known component of electronic circuits. It is essential to have good electrical isolation between p-type FET and n-type FET in a homogeneous CMOS structure. Some tuning methods can be used in TMD to achieve bipolar carrier conduction, such as work function engineering, chemical doping, electrostatic doping, and laser doping [19,20,21]. It is generally agreed that doping electronic devices with a variety of chemicals, which are not stable and reversible, reduces their operational functions. Therefore, for TMD based complementary devices, it is required to integrate n- and p-type materials-based FET [22]. The lack of control over the lateral dimensions, random thickness, and unexpected structural changes of few-layered MoTe_2_ caused by laser doping has made rigorously studying their intrinsic properties difficult [21,23,24]. Therefore, it is necessary to control p- and n-type laser doping in MoTe_2_-based FET by dielectric encapsulation materials, such as hexagonal boron nitride (hBN). Moreover, TMD properties and device applications can be clearly understood when air-stable and efficient surface doping materials are sought [25]. A heterogeneous type of 2D CMOS inverter has been predominantly reported, using two nanosheets for the p- and n-channels [22,26]. There have only been a few reports on homogeneous CMOS inverters using one flake to represent two distinct n- and p-type channels [27]. 

In this study, we report a laser irradiation-induced p-type doping, and through that we selectively converted hBN-encapsulated edge contact n-channel MoTe_2_ into p-channel MoTe_2_. The MoTe_2_ selective p-type doping technique enables the fabrication p-n homojunction with great potential for nanoelectronics applications. Therefore, to construct a p-n homojunction device in-plane with MoTe_2_, the laser-scanned region is p-doped. In contrast, the neighboring region is electrically n-doped when the back-gate voltage is positive. The laser doping technique is easier for realizing the in-plane MoTe_2_ p-n homojunction than other complex heterostructures. Furthermore, our laser-induced p-type doping technique has also been used to fabricate a CMOS inverter in a single MoTe_2_ nanoflake. In contrast to complex heterostructures, our simplified structure can be applied to fabricate 2D materials-based functional devices, such as photovoltaics and CMOS integrated circuits.

## 2. Materials and Methods

N-type 2H-MoTe_2_ single-crystals were grown by the self-flux method using our own optimized recipe. In brief, a vacuum-sealed quartz ampoule with source materials molybdenum powder of purity 99.99% (Sigma Aldrich, St. Louis, MO, USA) and tellurium 99.999% (Alfa Aesar, Haverhill, MA, USA) was measured and added into an alumina crucible. The tellurium material was used as a self-flux material and a reactive agent. The flux material has a lower melting point and aids dissolving reactive materials into a solution for the chemical reaction. The slow and controlled cooling rate of solution turns into crystallization process. The ampoule was vacuumed and sealed in an argon gas environment. Later it was placed inside the furnace for a few days and the temperature controller was allowed to cool down the furnace from 1100 °C to 550 °C with a cooling rate of 2.5 °C/h [10]. The ampoule was centrifuged and broken, and single 2H-MoTe_2_ crystals were harvested. In order to create the outer electrode pattern, a highly p-doped 285 nm Si/SiO_2_ substrate was photolithographically treated, and the In/Au metal layers were then deposited by an electron beam evaporator and removed with acetone. A mechanically exfoliated layer of hBN was transferred using the traditional Polydimethylsiloxane (PDMS) stamp method to reduce scattering and leakage currents [28]. After cleaning, the sample was submerged in chloroform, cleaned with acetone, and dried with a nitrogen gun. Next, another exfoliated hBN layer was picked onto the PDMS stamp. It is used to pick up exfoliated MoTe_2_ multilayer and dropped onto the bottom hBN in a patterned electrodes device. Afterward, methyl methacrylate was spin-coated onto the device and an e-beam lithography technique was used to draw inner electrodes on MoTe_2_. The sample was etched for edge contact and put for metal deposition, and (In/Au = 10:30 nm) was deposited onto the MoTe_2_ flake as top electrodes. Raman spectroscopy was performed by using a 532 nm laser in ambient conditions to characterize the flake. Electrical measurement was done under a vacuum in a dark environment using a 4155c analyzer.

## 3. Results and Discussion

Figure 1a illustrates schematic views of pristine and laser-scanned MoTe_2_ devices. Figure 1b shows the corresponding optical images of the pristine and laser-scanned MoTe_2_ devices. The red region shows laser-doped p-type MoTe_2_ FET, while blue corresponds to n-type MoTe_2_ FET. First, thick hBN was transferred on a precleaned Si/SiO_2_ 1 × 1 cm^2^ wafer to prevent the leakage current and interface traps. After that, a mechanically exfoliated multilayered flake of hBN was picked from the Si/SiO_2_ using a dry transfer method with a PDMS stamp. Later, it was used to pick up exfoliated multilayered MoTe_2_ flake and drop it onto the bottom hBN. Several electrodes (In/Au) were designed and fabricated on the top surface of the MoTe_2_ multilayered channel Appendix A. Through multiple laser scan cycles, MoTe_2_ could be continuously thinned. Laser scan cycles can be adjusted to achieve precise control of thickness. However, it is challenging, and hBN-encapsulation was needed for better controllability [23]. Raman spectroscopy is a convenient technique to characterize the fundamental properties of 2D materials [29]. In ambient conditions, Raman spectra of n-type MoTe_2_ flake were obtained using an excitation laser of 532 nm wavelength. Figure 1c illustrates the Raman spectrum of MoTe_2_ with three peaks: 171.2 cm^−1^ peak for $A_{1g}$, 232.2 cm^−1^ peak for $E_{2g}^{1}$, and 288.9 cm^−1^ peak for $B_{2g}$ [30].

The output characteristics (*I*_D_*-V*_D_) of the pristine and laser-scanned MoTe_2_ devices at *V*_G_ = 40 V are shown in Figure 2a,b, respectively. A linear dependency can be seen in the pristine n-type region of the MoTe_2_ FET device, indicating Ohmic contact, while the laser-doped MoTe_2_ FET exhibited Schottky-type contact. Figure 2c illustrates transfer characteristics (*I*_D_*-V*_G_) exhibiting n-type metal-oxide-semiconductor field effect transistor (MOSFET) behavior in pristine MoTe_2_ FET at a small drain voltage (*V*_D_) of 0.1 V. Figure 2d shows p-type MOSFET behavior at a drain voltage of 0.1 V in a laser-doped MoTe_2_ FET, indicating pristine n-type charge carrier polarity conversion into p-type charge carrier polarity. The laser-irradiated induced p-type doping in the MoTe_2_ channel originates from forming MoO_x_ layers on the top surface due to surface oxidation [25]. The MoO_x_ layer on the top surface efficiently traps electrons from the bottom layers of MoTe_2_ and thus contributes to p-type doping in MoTe_2_. The MoO_x_ has a high work function of about 6.6 eV and thus forms a rectifying contact with In in a p-doped FET [31]. This conversion of charge carrier polarity from n-type to p-type strongly depends on laser power and time. However, it is necessary to control local doping by adjusting the laser beam parameters [32]. To achieve controlled n-type to p-type conversion, we laser scanned different regions at constant laser power (optimized). In addition, to observe the consistency of FET behavior in laser-doped regions, we measured all devices at a range of temperatures (77–300 K), as shown in Figure 3.

Figure 3a shows transfer characteristics of pristine n-type MoTe_2_ FET at the temperature range (77–300 K) at a drain voltage (*V*_D_) of 0.1 V in sweeping gate biasing −40 V to 40 V, where output characteristics are shown in Appendix A. The device shows an intrinsic behavior and stable electron charge transport properties at all measured temperatures. Afterward, the device was laser scanned on the edge part for 2 min while keeping constant laser power of about 2.14 µW, and the device shows weak and strong ambipolar charge transport behavior at all ranges of temperatures (77–300 K), as shown in Figure 3b,c. After that, we laser-scanned the whole MoTe_2_ FET channel using the same time and power parameters as stated above (t = 2 min, P = 2.14 µW) and the device shows complete conversion from n-type to p-type charge carrier transport at 77 K to 300 K, as shown in Figure 3d. We refer n-to-p-type charge conversion in our device due to the formation of MoO_x_ layer on the surface of the MoTe_2_ channel caused by the surface oxidation, where it traps the electrons from the bottom layer and contributes to the hole transport carriers in the channel as stated earlier. Based on the charge carrier polarity conversion, each case charge transport mechanism and Fermi level modulation are drawn schematically, as shown in band alignment Figure 3e–h.

The field-effect mobility (${µ}_{\mathrm{FE}}$) of the MoTe_2_ device is calculated and plotted using ${\;µ}_{\mathrm{FE}}=g_{m}\times \left[\frac{L}{{\mathrm{WC}}_{\mathrm{ox}}V_{\mathrm{DS}}}\right]$ where $\;g_{m}=\left(dI_{\mathrm{DS}}/dV_{G}\right)$ is the transconductance, *L* is the channel length, *W* is the width, *C*_ox_ is the capacitance of 285 nm thick SiO_2_, *I*_D_ is the source-drain current, *V*_G_ is the gate voltage, and *V*_D_ represents the drain voltage [33]. The electron mobility at room temperature in the n-type MoTe_2_ FET is 23.4 cm^2^V^−1^s^−1^ at a small drain voltage (*V*_D_) of 0.1 V, while hole mobility at room temperature in the laser-doped p-type MoTe_2_ FET is 0.61 cm^2^V^−1^s^−1^ at 0.1 V. Both electron and hole mobilities show strong temperature dependence (Appendix A). Furthermore, our study also demonstrated the CMOS behavior of the MoTe_2_ device.

A complementary inverter is formed when n-type and p-type FET are combined after a controlled laser-doped n-to-p conversion. In general, n-type FET are grounded, and p-type FET are supplied with voltage (*V*_dd_). Regardless of the FET type, back gates are input voltages (*V*_IN_) for both p-type and n-type devices. The n-type and p-type electrodes are connected to measure the output voltage (*V*_OUT_). Figure 4b shows the CMOS inverter configuration with an illustration of the logic circuit. An illustration of the transfer characteristic of the inverter as a function of V_IN_ depicted in Figure 4c shows a sharp voltage transition with varying input voltages in the range of 0.5 to 2 V. Figure 4d shows a maximum voltage gain of 0.11 at *V*_dd_ = 2 V specified as gain = *dV*_OUT_/*dV*_IN_. Using high-k dielectrics can further improve this gain, supporting the potential of the MoTe_2_ CMOS technology based on a single channel.

## 4. Conclusions

In summary, we have demonstrated the laser-irradiated technique for the p-type doping of a multilayer hBN-encapsulated MoTe_2_ device. The p-type doping in MoTe_2_ arises from an overlayer of high-work function-oxidized MoO_x_ over the laser-irradiated region. It is possible to selectively dopped and switch a carrier’s polarity by carefully controlling laser power or time in an encapsulated hBN MoTe_2_ device while keeping neighboring regions intact. In this way, a positive gate voltage combined with laser irradiation can create an in-plane p-n heterojunction. Thus, this study demonstrates the great potential of selective doping in hBN-encapsulated MoTe_2_-based FET by using laser scans and shows its potential use in the fabrication of CMOS circuits.

## Figures and Tables

**Figure 1 nanomaterials-13-01700-f001:**
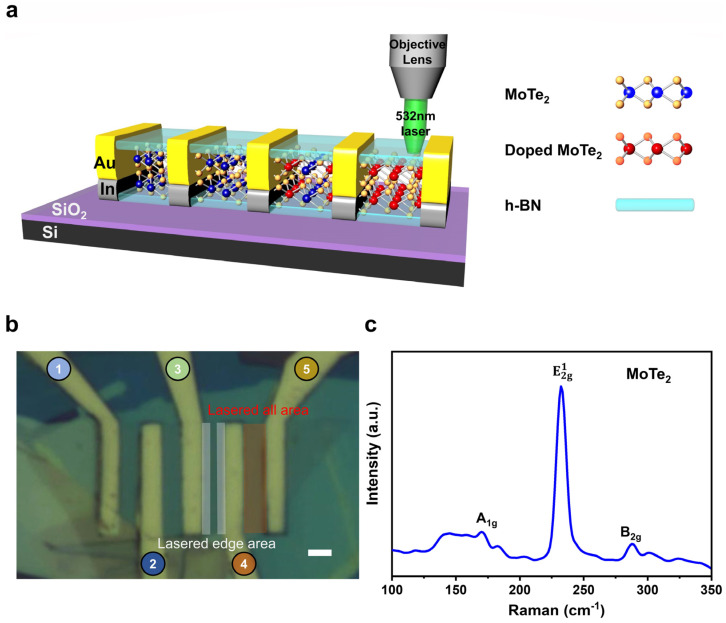
Schematic and optical representation of MoTe_2_ FET. (**a**) Schematic diagram of the MoTe_2_ FET device. (**b**) Optical image of the MoTe_2_ FET device with In metal contacts (scale bar 2 µm). (**c**) Raman spectrum of few layers intrinsic MoTe_2_ FET device.

**Figure 2 nanomaterials-13-01700-f002:**
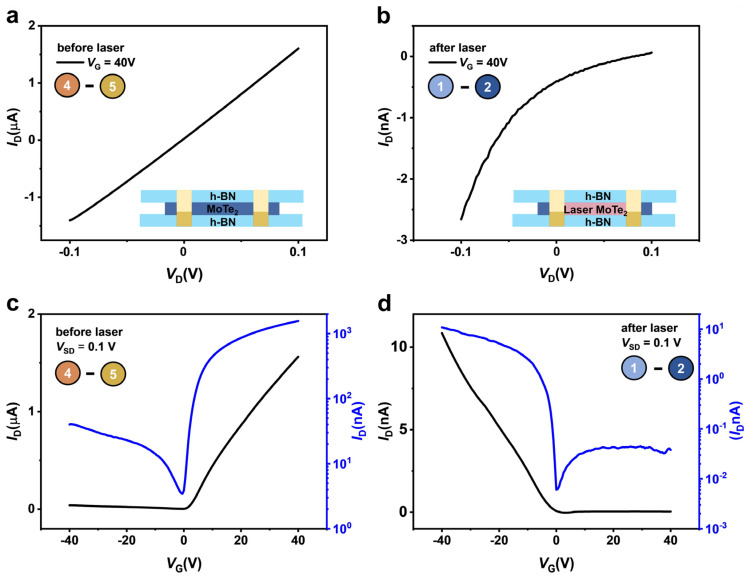
Plots comparing the output and transfer characteristics of different electrodes. (**a**) *I*_D_*-V*_D_ characteristics of MoTe_2_ channel before laser scanning. (**b**) *I*_D_*-V*_D_ characteristics of MoTe_2_ channel after laser scanning. (**c**) *I*_D_*-V*_G_ characteristics plot of MoTe_2_ channel before laser scanning. (**d**) *I*_D_*-V*_G_ characteristics plot of MoTe_2_ channel after laser scanning.

**Figure 3 nanomaterials-13-01700-f003:**
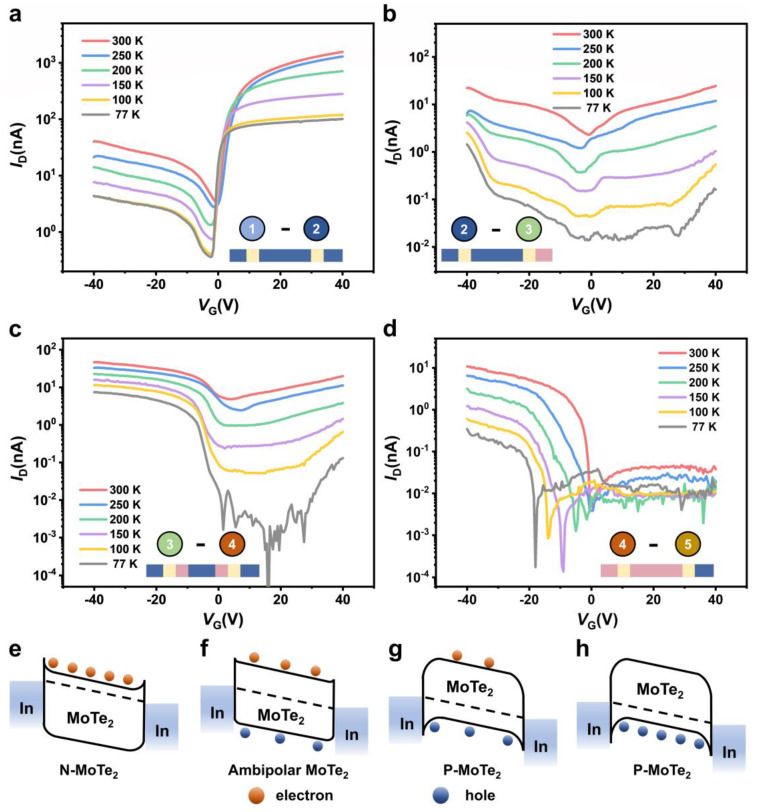
Transfer characteristics and Band alignment of the MoTe_2_ before and after laser scanning. (**a**) *I*_D_*-V*_G_ characteristics plot of MoTe_2_ channel (light blue) before laser scanning. (**b**) *I*_D_*-V*_G_ characteristics plot of MoTe_2_ channel after laser scanning on c part edges. (**c**) *I*_D_*-V*_G_ characteristics plot of MoTe_2_ channel after laser scanning on edge parts of the channel. (**d**) *I*_D_*-V*_G_ characteristics plot of MoTe_2_ channel (pink) after laser scanning on the whole channel. (**e**–**h**) Band alignment of the MoTe_2_ before and after laser scanning.

**Figure 4 nanomaterials-13-01700-f004:**
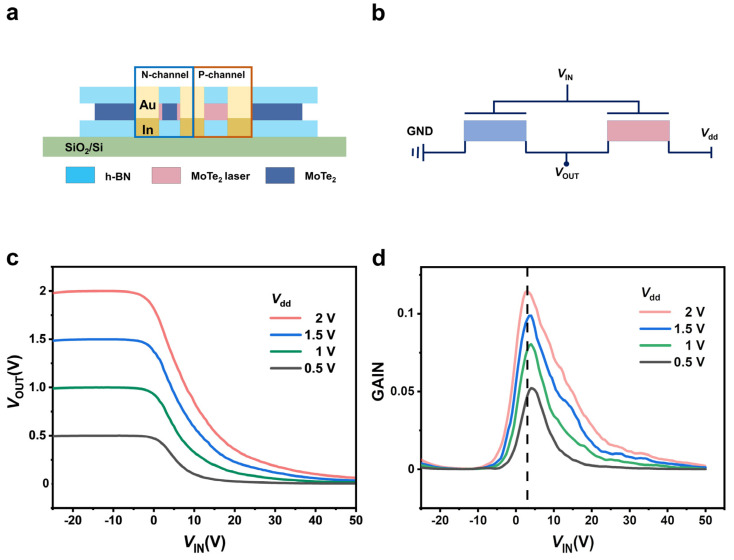
MoTe_2_ FET-based inverter. (**a**) Blue and orange line boxes indicate the n-type and p-type FET in the MoTe_2_-based CMOS inverter schematic. (**b**) MoTe_2_-based CMOS inverter circuit diagram. (**c**) A plot of the voltage transfer characteristics of the inverter as a function of *V*_IN_ at several *V*_dd_. (**d**) The voltage gain of an inverter is a function of supply voltages.

## Data Availability

Not applicable.

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
