# Peer review of "Conversion of Charge Carrier Polarity in MoTe2 Field Effect Transistor via Laser Doping"

_nanomaterials, 2023, doi:10.3390/nano13101700_

Round 1
Reviewer 1 Report
The manuscript demonstrated laser-induced p-type doping in a selective region of n-type semiconducting MoTe2 field effect transistors (FET) and further revealed an inverter. The absorption of atomic-thin 2D layers irradiated by the green laser should be quite low and damage on passivated interfaces likely was barely negligible or at least a concern. Therefore, giving the detailed laser power dependence of structural change, device performance was helpful and mechanisms on charge carrier polarity due to laser-doping was also mandatory. Moreover, significantly low mobility and on-currents and even degraded sub-threshold slope on laser-enabled p-FETs maybe reflects to the limitation for further developing such technologies.
The text was well written.
Reviewer 2 Report
Kim et al. report the p-doping of MoTe2-based field effect transistors by using laser irradiation methods. Authors have performed detailed characterization that includes Raman spectroscopy and I-V characteristics. The manuscript is well written, but some grammatical improvements can be made. Conclusions are consistent with the experimental results. I would recommend publication to Nanomaterials after some minor revision:
1. Please provide more details about the synthesis of the single crystals, so that the experimental section is more complete. You could keep reference #10 for the synthesis.
2. Please read proof the manuscript one more time. For example, lines 26-27 in the abstract need re-writing.
Please read proof the manuscript one more time. For example, lines 26-27 in the abstract need re-writing.
Reviewer 3 Report

Needs to review a few sentences mentioned in the comments.
Reviewer 4 Report
Comments
The authors demonstrated laser-doping induced the change of charge carrier polarity in MoTe2-based FET devices. They also showed an inverter device based on the integrated N- and laser-induced P-type devices. Clear evidence of the conversion from N-type to P-type doping can be concluded based on the electrical performance. However, some key issues should be addressed before the manuscript could be considered for publication.
1. The authors claimed the change from n-to p-type charge conversion occurs because of the formation of the MoOx layer on the surface. I am wondering if there are any material characterization results that support this claim. For example, Is there any difference in Raman spectra for n-type and p-type MoTe2 flake?
2. What is the thickness of the MoTe2 flake for the device? Is it possible that the laser scan could induce the thinning of the MoTe2 flake as this has been reported in many reports in laser-2D material interaction? More information on the characterization of the flake before and after the laser scanning experiment is required.
3. Information on the laser scan experiments is missing in the manuscript. For example, laser type, wavelength, scan parameter, etc. This is important for other researchers to reproduce the results as evidenced in this manuscript.
4. The role of h-BN on the device performance is not clearly described in the manuscript. I am wondering if the material property of the h-BN flake has changed before and after laser scanning.
5. Some more discussion on Figure 3(e-h) is recommended to make a better understanding. It might be better to put each of these figures as inset figures in (a-d)?
Some clear grammar errors should be carefully corrected throughout the manuscript. For example, In the abstract, the sentence of “in an intrinsic n-type channel and hole mobility of about 0.61 cm2V-1s-1 with a high on/off 25 ratio which has measured the device in temperature dependence makes better understanding in a laser-doped channel” is confusing and should be further improved. In Line 60, “complementary devices based on are TMD to be built by integrating n- and p-type FET” should be “complementary devices based on TMD are to be built by integrating n- and p-type FET”. In line 165, “we laser-scanned the whole MoTe2 FET time of 2min and power (2.14uW)” is also confusing.
Round 2
Reviewer 1 Report
The revised version could be published.
That's fine.